

# Whole-genome amplification in double-digest RADseq results in adequate libraries but fewer sequenced loci

Bruno A. S. de Medeiros and Brian D. Farrell

Department of Organismic and Evolutionary Biology and Museum of Comparative Zoology, Harvard University, Cambridge, MA, USA

## ABSTRACT

Whole-genome amplification by multiple displacement amplification (MDA) is a promising technique to enable the use of samples with only limited amount of DNA for the construction of RAD-seq libraries. Previous work has shown that, when the amount of DNA used in the MDA reaction is large, double-digest RAD-seq (ddRAD) libraries prepared with amplified genomic DNA result in data that are indistinguishable from libraries prepared directly from genomic DNA. Based on this observation, here we evaluate the quality of ddRAD libraries prepared from MDA-amplified genomic DNA when the amount of input genomic DNA and the coverage obtained for samples is variable. By simultaneously preparing libraries for five species of weevils (Coleoptera, Curculionidae), we also evaluate the likelihood that potential contaminants will be encountered in the assembled dataset. Overall, our results indicate that MDA may not be able to rescue all samples with small amounts of DNA, but it does produce ddRAD libraries adequate for studies of phylogeography and population genetics even when conditions are not optimal. We find that MDA makes it harder to predict the number of loci that will be obtained for a given sequencing effort, with some samples behaving like traditional libraries and others yielding fewer loci than expected. This seems to be caused both by stochastic and deterministic effects during amplification. Further, the reduction in loci is stronger in libraries with lower amounts of template DNA for the MDA reaction. Even though a few samples exhibit substantial levels of contamination in raw reads, the effect is very small in the final dataset, suggesting that filters imposed during dataset assembly are important in removing contamination. Importantly, samples with strong signs of contamination and biases in heterozygosity were also those with fewer loci shared in the final dataset, suggesting that stringent filtering of samples with significant amounts of missing data is important when assembling data derived from MDA-amplified genomic DNA. Overall, we find that the combination of MDA and ddRAD results in high-quality datasets for population genetics as long as the sequence data is properly filtered during assembly.

Corresponding author
Bruno A. S. de Medeiros, souzademedeiros@fas.harvard.edu

## INTRODUCTION

Double-digest RAD sequencing (ddRAD) (*Peterson et al., 2012*) and other methods of genotyping-by-sequencing are inexpensive and flexible tools that allow researchers to sequence a large number of loci from non-model organisms for phylogenetic and population-level studies (*Andrews et al., 2016*). In spite of its many advantages, a potential barrier to the use of ddRAD is the relatively large amount of starting DNA typically required in library preparation. This may hinder its application for DNA obtained from museum specimens, rare species (*Blair, Campbell & Yoder, 2015*), or small organisms (*Shortt et al., 2017*; *Boyle et al., 2018*), which yield limited amounts of DNA. A possible solution consists in preparing RAD libraries from pools of individuals, which together yield sufficient DNA (*Emerson et al., 2010*; *Toonen et al., 2013*). Pooling offers the additional benefit of reducing library preparation costs (*Toonen et al., 2013*; *Schlötterer et al., 2014*). However, accurate genotyping of pools is challenging (*Cutler & Jensen, 2010*; *Gautier et al., 2013*; *Lynch et al., 2014*) and might require impractically high number of samples pooled at equimolar concentrations (*Cutler & Jensen, 2010*; *Lynch et al., 2014*). An alternative to pooling is to prepare libraries with methods that require smaller DNA input. One of these alternatives is HyRAD (*Suchan et al., 2016*; *Linck et al., 2017*), the hybrid enrichment of genomic libraries for RAD-derived loci. One caveat is that this protocol is still not highly optimized, requiring a very large read depth in comparison to ddRAD and thus resulting in a much larger cost per sample in addition to the higher cost for library preparation. Another option for samples with smaller DNA amounts is to increase the DNA available per individual by using whole-genome amplification prior to RAD library preparation. In the context of ddRAD, *Blair, Campbell & Yoder (2015)* sequenced four samples of a single species at high coverage and using a high amount of starting DNA, following the manufacturer's protocol for the reaction of whole-genome amplification. Even though their results were encouraging, it remains to be shown whether whole-genome amplification is also robust in more typical conditions in which it might be used: the study of many samples with uneven coverage and small quantities of DNA available.

The method of whole-genome amplification used in this previous study, multiple displacement amplification (MDA) (*Dean et al., 2002*; *Hosono et al., 2003*), is arguably one of the best among those available in terms of detectable biases in the amplified product (*Sabina & Leamon, 2015*). MDA consists of isothermal amplification of DNA using random primers for an extended time. For the applications of whole-genome sequencing and PCR, MDA has exhibited no genotyping bias and even coverage of different genomic regions under most conditions, although some studies have reported a GC-dependent coverage bias (*Sabina & Leamon, 2015*). When applied to ddRAD libraries, MDA resulted in accurate genotype calls, with no detectable coverage bias (*Blair, Campbell & Yoder, 2015*). These results were based on a few samples sequenced to high depth with a fairly large amount of template DNA (>100 ng). Typical population-level studies in which MDA could be most useful would include those in organisms with significantly smaller DNA yields, and possibly pools of many samples with uneven coverage.

For example, *Boyle et al. (2018)* have used MDA in ants to obtain enough DNA (150 ng) to start ddRAD library preparation from an initial template with less than 15 ng of DNA (J. Boyle, 2018, personal communication). With even smaller extraction yields, *Shortt et al. (2017)* have used MDA to prepare libraries from miracidia larvae of the flatworm *Schisosoma japonicum*, which typically yield one to two ng of genomic DNA.

An important concern about whole-genome amplification, not addressed in previous studies focusing on its application to ddRAD, is non-templated amplification. MDA reactions are known to amplify contaminants present in the MDA reagents (*Sabina & Leamon, 2015*), and extraneous DNA present in the genomic DNA or the reaction setup might also be amplified and detectable in the final product. Even though any kind of ddRAD preparation starting from low amounts of DNA might be susceptible to some level of contamination, MDA may, in principle, increase the amount of contaminants in the sequenced library. While some level of non-templated amplification can be removed by filtering out loci not shared across multiple samples during dataset assembly, the effects of non-templated amplification on the final dataset have not been tested so far.

Insects are one group of small animals for which MDA might be useful. Some small insects yield little DNA from extractions, and in these cases destructive extraction from whole bodies is needed. In our experience working with small beetles, even extractions from whole bodies result in yields from a few nanograms to a few micrograms of DNA. Since the lower end of these yields is also in the lower end of the input DNA required for the preparation of RAD libraries (*Andrews et al., 2016*), MDA may enable the use of samples that yield little DNA. Here, we study the effects of MDA on ddRAD libraries prepared for a number of species of small (ca. three to seven mm in length) weevils (Insecta, Coleoptera, Curculionidae). Taking advantage of the natural variation in DNA yield across samples, we evaluate the effect of varying amounts of input genomic DNA into an MDA reaction. By comparing libraries generated with and without MDA from the same genomic DNA, we also evaluate the level of non-templated amplification, and the effect of MDA on heterozygosity, genotyping bias and on the loci obtained for a given sequencing effort. We also test the use of a mix of amplified and non-amplified samples in common population genetics analyses: population clustering and isolation by distance.

## MATERIALS AND METHODS

### Sampling, library preparation, and sequencing

Here, we work with five taxa of weevils associated with the flowers of the palms *Syagrus coronata* and *S. botryophora*: *Anchylorhynchus trapezicollis*, *Andranthobius bondari*, *Celetes impar*, *Microstrates bondari*, and *M. ypsilon*. Our preliminary analyses using a larger dataset showed evidence for cryptic speciation in *Anchylorhynchus trapezicollis* and *Andranthobius bondari*. Here, we will refer to these taxa here using only the generic name, but we still include all samples assigned to these species based on morphology alone. Weevils were collected throughout the range of their host palms between 2013 and 2014 by bagging inflorescences in the staminate phase. Insects were killed and preserved in 96% ethanol and kept in a cooler and then a freezer prior to DNA extraction. All samples

studied here were collected with SISBIO permit number 39704-7 from the Instituto Chico Mendes de Preservação da Biodiversidade, Brazil, and deposited in the Museum of Zoology of the University of São Paulo. Some individuals in each species had ddRAD libraries prepared both with and without an MDA step (here referred as MDA and gDNA libraries, respectively), and we also included in this study individuals with only one library prepared, either with or without MDA, that were collected in the same localities as the individuals with two libraries. This resulted in 69 samples with the two kinds of library prepared, 76 samples with a gDNA library only and 22 samples with an MDA library only. Sample data are listed in Table S1 and a graphical overview of the samples included in this study is given in Fig. 1.

For gDNA libraries, we have followed the protocol outlined by *Peterson et al. (2012)*, but using MagNA beads (*Rohland & Reich, 2012*) instead of Ampure XP beads (Beckman Coulter, Brea, CA, USA) for cleanup steps. DNA was extracted either automatically in an AutoGenprep 965 (AutoGen, Holliston, MA, USA) or using the EZNA Insect DNA Kit (Omega Biotek, Norcross, GA, USA). We extracted DNA from whole bodies separated between the elytra and pronotum, to preserve the remaining sclerotized tissue (the head, prothorax and appendages) as vouchers, and digests were done overnight prior to extraction. DNA was quantified using a Quant-iT dsDNA HS Assay Kit (Thermo Fisher, Waltham, MA, USA) in a Spectramax i3 plate reader (Molecular Devices, San Jose, CA, USA). A total of 150–300 ng of DNA was digested using the restriction enzymes EcoRI-HF and BfaI (New England Biolabs, Ipswich, MA, USA), cleaned with magnetic beads and quantified. A total of 50–100 ng of digested DNA was used in a ligation reaction to add unique five-bp inline barcodes with a T4 DNA Ligase (New England Biolabs, Ipswich, MA, USA). After adapter ligation, we prepared pools of 16–48 uniquely barcoded samples, which were then cleaned with magnetic beads and size-selected in the range of 264–336 bp in a Pippin Prep (Sage Science, Beverly, MA, USA). Each pool was PCR-amplified to increase DNA amount and to add an Illumina P5 barcode using a Phusion High-Fidelity PCR Kit (New England Biolabs, Ipswich, MA, USA), with four to eight separate reactions ran in parallel for each pool, depending on the volume obtained after size selection. We used the following program for PCR reactions: initial denaturation at 98 °C for one min, 10–12 cycles of denaturation at 98 °C for 10 s, annealing at 65 °C for 30 s and extension at 72 °C for 30 s, followed by a final extension at 72 °C for seven min. PCR cycles were limited to 12 to avoid amplification biases (*DaCosta & Sorenson, 2014*). PCR products from a single pool were pooled and cleaned together using MagNA beads. The size distribution of final libraries was inspected in an Agilent Bioanalyzer 2100 or Agilent Tapestation 2200. Libraries were quantified using a Qubit dsDNA HS Assay Kit (Thermo Fisher, Waltham, MA, USA) in a Qubit 2.0 Fluorometer (Thermo Fisher, Waltham, MA, USA) and pooled for sequencing considering their DNA concentrations and number of samples in each pool, with the aim of achieving an equal number of reads for all samples. Libraries were sequenced and Illumina barcodes demultiplexed by the Bauer Core Facility at Harvard University. Pools were spread throughout multiple sequencing runs to be multiplexed with unrelated more diverse libraries (usually RNA-seq) and avoid

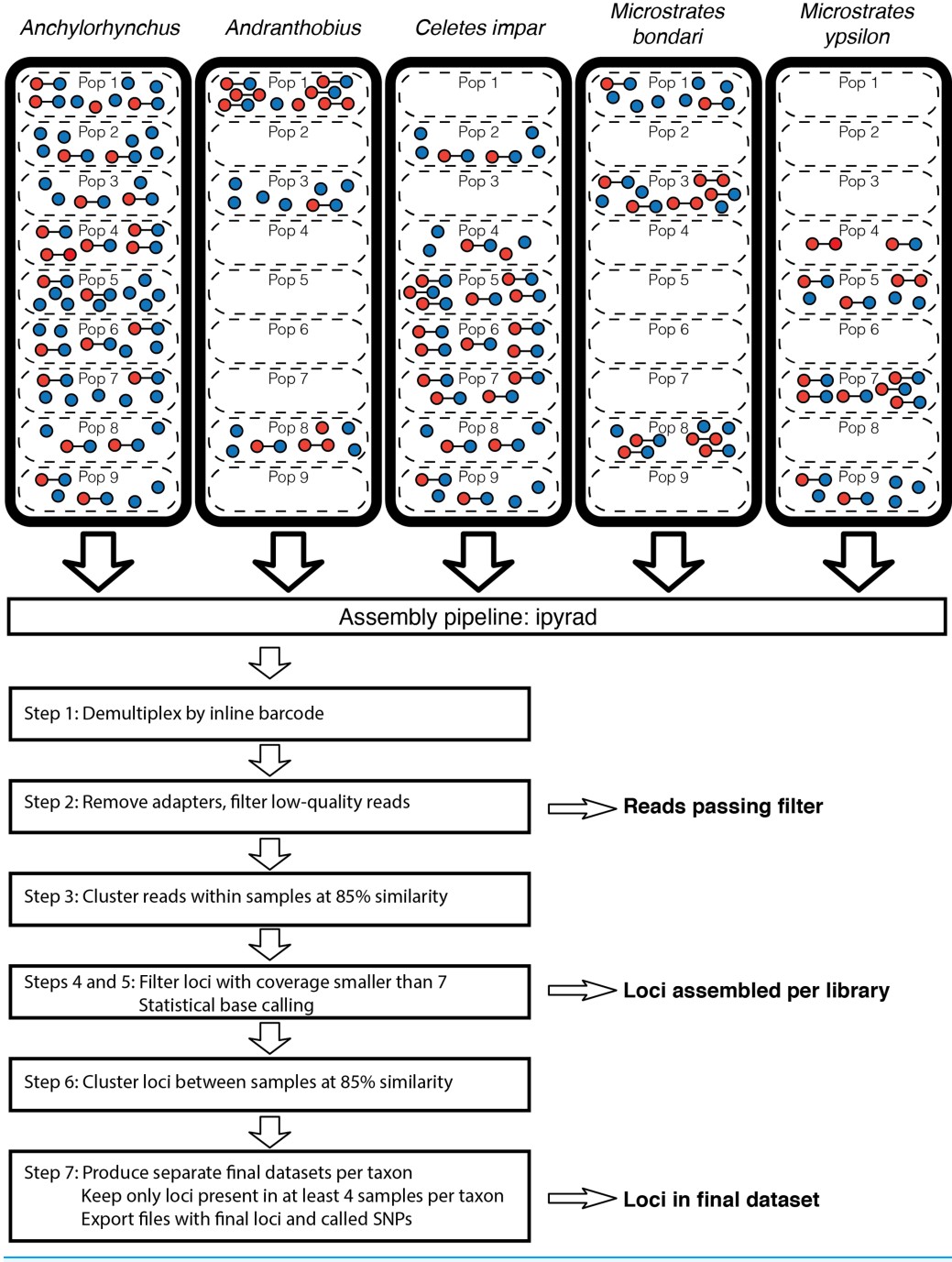

**Figure 1  Overview of samples included in the study and the ipyrad pipeline for dataset assembly.** For each taxon, dashed rectangles represent populations and each circle represent one ddRAD library. Libraries prepared from the same individual are linked by a line. Red circles represents MDA libraries and blue represents gDNA libraries. An overview of the ipyrad assembly pipeline used to generate independent datasets for each taxon is given in the lower half of the figure and arrows indicate steps from which sequences were retrieved to study the effects of MDA. The same terminology is used throughout the text to refer to sequences at these steps. Locality data for populations is available in Table S1.

problems with base calling caused by the low diversity of restriction sites and inline barcodes. The minimum read length of these runs was single-end 100 bp, and all sequences with longer reads were trimmed to this length prior to assembly.

For MDA libraries, 1.5–2 μL (6–360 ng) of extracted DNA was amplified for 16 h at 30 °C followed by three min at 65 °C using a Repli-G Mini Kit (Qiagen, Germantown, MD, USA) in 15–20 μL reactions (protocol in Article S1). We have performed reactions in smaller volumes than the recommended manufacturer protocol (50 μL) because smaller reaction volumes seem to reduce biases in MDA for smaller DNA quantities (*Sabina & Leamon, 2015*), and also result in a reduced cost per sample. Following amplification, DNA was quantified with a Quant-It assay and directly used in restriction digests, with the remaining library preparation following the gDNA protocol.

### Bioinformatics

Sequences were demultiplexed by inline barcodes and assembled using ipyrad v. 0.6.8 (*Eaton, 2014*; *Eaton & Overcast, 2017*). An overview of the steps in the ipyrad pipeline is given in Fig. 1. No mismatches were allowed in barcodes during demultiplexing, and bases with a quality score lower than 20 or containing adapters were trimmed from the 3′ end. After trimming, reads with less than 35 bp or with five or more low-quality bases were removed. Reads were clustered both within and between samples at 85% identity, and only loci with coverage ≥7 were retained for statistical base calling. In the ipyrad pipeline, genotypes are statistically called independently for each sample for sites above the minimum coverage. Throughout this paper, we use the phrase "assembled loci" to refer to this set of loci assembled independently for each sample and filtered for minimum coverage. For assembly of the final dataset, we set, for each species, the minimum number of samples for a locus to four, and we will refer to these loci as "loci in the final dataset." Input files for ipyrad are provided in Data S1. MDA and gDNA libraries from a single sample were treated as different samples during assembly. Final datasets were generated independently for each species by using the "branch" option in ipyrad.

### Effect of starting DNA amount on the number of reads obtained for MDA libraries

To test whether MDA impacts the number of loci obtained, we counted, for all MDA libraries (including those that yielded few loci or that do not have a corresponding gDNA library), the number of reads obtained and number of loci assembled. We modeled the effect of input genomic DNA used in the MDA reaction by using a linear mixed model with number of loci as the response variable, mass of input genomic DNA and number of reads as fixed effects, and sequencing pool and taxon as random effects affecting the intercept. We started by fitting a full model including all fixed effects and their interactions and simplified the model using the R function step to remove non-significant fixed effects stepwise using *F*-tests with significance level 0.05. We adopted this same procedure when fitting all mixed models throughout this study. Models were fit using the R packages lme4 v. 1.1–12 (*Bates et al., 2015*) and lmerTest v. 2.0–33 (*Kuznetsova, Brockhoff & Christensen, 2017*) in R v. 3.3.2 (*R Core Team, 2016*).

## Comparisons between MDA and gDNA libraries for loci assembled, variation in coverage, GC content, and heterozygosity

To characterize other possible effects of MDA, we compared several statistics between gDNA and MDA libraries. These comparisons were done only in a subset of the samples passing a minimum filter of number of loci shared between samples, to ensure that the calculated statistics were a reasonable representation of the whole set of genes potentially sequenced with ddRAD. The number of reads per sample in a pool were extremely variable. This variation is a direct result of the study design: the samples studied here are a subset of samples sequenced primarily for a phylogeographic study, and some of the re-sequenced samples had resulted in few reads during the first attempt. To overcome this limitation, we only compare gDNA and MDA libraries that resulted in at least 100 assembled loci and we use number of reads or number of loci as a covariate in all analyses. Moreover, we filtered additional samples based on the results from the analysis on non-templated amplification (see below). These showed that all samples with evidence for contamination are among those with fewer loci shared with other samples in the final dataset. To avoid any biases caused by contamination, and not MDA per se, we excluded the nine MDA libraries with smallest number of loci in the final dataset for the paired comparisons in this section. In total from the 67 pairs of libraries prepared, 48 were retained after both filters. Table S1 and Table 1 include information about which samples were included or excluded from each model.

Following sample filtering, we tested whether MDA libraries result in fewer loci than their gDNA counterparts by fitting a linear mixed model with MDA and number of reads as fixed effects, and sample within taxon as random effects (these same random effects were used in all mixed models comparing gDNA and MDA libraries). In order to achieve approximate normality of residuals, we logit-transformed the response variable, assuming a maximum of 30,000 loci for a sample. We tested if MDA increases the variation in coverage across loci by fitting a model with the standard deviation of coverage in assembled loci as a response and number of reads and MDA as fixed effects.

We also tested the effect of MDA on heterozygosity and GC content by fitting a linear mixed model with the same random effects as above, and MDA and number of assembled loci as fixed effects. In the case of heterozygosity, we also fitted a model with MDA and average coverage per assembled locus as fixed effects. Heterozygosity was calculated as observed heterozygosity: the proportion of heterozygous sites for each library in loci retained in the final dataset, after base calling. In all cases, if plots of residuals vs. predicted values and quantile-quantile plots showed strong deviation from normality, we log-transformed the response variable or some of the fixed effects. All linear mixed models fitted and data transformations are listed in Table 1.

## Biased recovery of loci under MDA

After finding that MDA results in fewer sequenced loci, we tested whether this is due to consistent amplification of the same set of loci or due to stochastic effects. If deterministic effects predominate, it is expected that the set of loci shared by two samples should be more similar between MDA libraries or between gDNA libraries than

**Table 1 Results of mixed model fitting, with sample inclusion criteria.**

| Response | Samples included | Fixed effects | Estimate (*p*-value) |
|---|---|---|---|
| **Number of assembled loci** | All MDA libraries | Intercept | −6,624.93 (0.315) |
| | | Reads passing filter (log) | 1,061.32 (0.036) |
| | | Input gDNA in MDA reaction (ng) | −227.42 (<0.001) |
| | | Interaction | 18.213 (<0.001) |
| Proportion of reads matching to incorrect taxa (logit) | 57 samples with both kinds of library and at least 100 loci in final dataset | Intercept | −2.83 (0.034) |
| | | Reads passing filter (log) | −0.25 (0.012) |
| | | MDA | |
| | | Interaction | |
| Proportion of assembled loci matching to incorrect taxa (logit) | 57 samples with both kinds of library and at least 100 loci in final dataset | Intercept | −6.30 (<0.001) |
| | | Loci with coverage ≥7 (log) | |
| | | MDA | |
| | | Interaction | |
| **Proportion of loci in final dataset matching to incorrect taxa (logit)** | 57 samples with both kinds of library and at least 100 loci in final dataset | Intercept | −6.47 (>0.001) |
| | | Loci in final dataset (log) | 0.003 (0.97) |
| | | MDA | 3.98 (<0.001) |
| | | Interaction | −0.45 (<0.001) |
| **Number of assembled loci (logit)** | 48 samples with both kinds of library and largest number of loci in final dataset | Intercept | −19.97 (<0.001) |
| | | Number of reads (log) | 3.55 (<0.001) |
| | | MDA | 4.73 (0.040) |
| | | Interaction | −0.99 (0.016) |
| **Standard deviation of read depth across loci** | 48 samples with both kinds of library and largest number of loci in final dataset | Intercept | −2.73 (0.007) |
| | | Number of reads (log) | 1.18 (<0.001) |
| | | MDA | 0.89 (<0.001) |
| | | Interaction | |
| **Heterozygosity** | 48 samples with both kinds of library and largest number of loci in final dataset | Intercept | $-7.66 \times 10^{-4}$ (0.53) |
| | | Number of assembled loci (log) | $5.65 \times 10^{-4}$ (<0.001) |
| | | MDA | $-7.17 \times 10^{-3}$ (0.012) |
| | | Interaction | $7.69 \times 10^{-4}$ (0.016) |
| **Heterozygosity** | 48 samples with both kinds of library and largest number of loci in final dataset | Intercept | $5.73 \times 10^{-4}$ (0.53) |
| | | Average coverage across assembled loci (log) | $1.09 \times 10^{-3}$ (<0.001) |
| | | MDA | $7.99 \times 10^{-3}$ (0.012) |
| | | Interaction | $-2.21 \times 10^{-3}$ (0.016) |
| %GC | 48 samples with both kinds of library and largest number of loci in final dataset | Intercept | 0.34 (<0.001) |
| | | Number of assembled loci (log) | $-1.25 \times 10^{-3}$ (<0.001) |
| | | MDA | |
| | | Interaction | |

**Notes:**
Data transformations, if any, are indicated in parenthesis. Estimates and *p*-values are shown only for effects retained in the final models. Models in which MDA had a significant effect have their response highlighted in bold (including the first model, in which all included data came only from MDA libraries). Code and complete model summaries can be found in Article S2.

when comparing between both kinds of libraries. Libraries produced from more closely related individuals are also expected to result in a more similar set of loci due to higher similarity in restriction sites, and libraries that were pooled together for size selection and PCR are also expected to be more similar due to more consistent size selection. Therefore, to assess whether the set of loci produced by MDA is biased, we have implemented a model that jointly infers the effect of all of these variables in the pairwise dissimilarity in the set of loci recovered for samples. For each taxon, we calculated the pairwise dissimilarity in the final dataset as 1-$p$, where $p$ in the proportion of final loci shared for both libraries. All libraries were included in this analysis, without any filter, since even libraries that do not share any locus provide relevant information. We modeled the relationship between this dissimilarity matrix as a response and several predictors that we believe could also be associated with the recovery of a more similar set of final loci (MDA, population (i.e., locality), size selection pool and log-transformed number of loci in the final dataset for each sample) using a multivariate distance matrix regression (MDMR) (*Anderson, 2001*; *Mcardle & Anderson, 2001*) implemented in the R package MDMR v. 0.5.0 (*McArtor, Lubke & Bergeman, 2016*; *McArtor, 2017*, *2018*). This kind of regression tests the relationship between several variables and a distance matrix, drawing significance from randomization. Here significant relationships indicate that similar values of a given predictor are associated with libraries having more similar sets of loci in the final dataset.

## Non-templated amplification

To assess the effect of non-templated amplification caused by MDA, we aligned reads and loci obtained for each library to a reference dataset. Since no genomes are available for the species studied here or for any close relatives, we produced references from the final alignments generated by ipyrad. For each taxon, we combined in a fasta file all sequences from gDNA libraries found in the final dataset. We included only loci from the final alignment in the reference data because the filters used in ipyrad (particularly the minimum of four samples in an alignment) should minimize the presence of contaminants in the reference set, at the cost of making it less comprehensive. These sequences were de-replicated using cd-hit (*Li & Godzik, 2006*; *Fu et al., 2012*) at 95% similarity level and the resulting sequences for all species were combined in a single fasta file. To assess contamination from human DNA, we also added to this file the repeat-masked Human Genome Assembly GRCh38.p10, downloaded from NCBI, with repeats (i.e., lower case letters) replaced by Ns prior to concatenation.

This reference fasta file was used to generate a bowtie2 (*Langmead & Salzberg, 2012*) database. We then aligned, for all samples with both gDNA and MDA libraries, sequences to this reference at three different steps during dataset assembly: adapter-trimmed reads, all loci assembled for each sample with minimum coverage of seven, and all loci for each sample that were retained in the final dataset (i.e., were shared by at least another four samples). All alignments were made with the "very-sensitive" preset (which is equivalent to using options -D 15 -R 2 -L 22 -i S,1,1.15). A custom bash script was used to

parse the SAM files produced by bowtie2 and count the number of reads and loci aligning to each reference species, as well the number of unaligned sequences. Since the reference sequences are non-comprehensive and based on gDNA libraries themselves, it is expected that MDA libraries will have a smaller proportion of correctly aligned sequences and a larger proportion of unaligned sequences. Due to bowtie's heuristic algorithm, it is also possible that a small proportion of sequences might align to incorrect taxa both in gDNA and MDA libraries, especially in the case of closely related species (for species in different genera, our ddRAD datasets have almost no homologous loci). However, an increase in the number of incorrect alignments under MDA is only expected in the case of significant non-templated amplification. To test whether this is observed here, we modelled the effect of MDA on the number of reads/loci aligning to incorrect taxa by using the logit-transformed proportion of incorrect matches as the response variable in a mixed model with MDA and log-transformed number of reads/loci as fixed effects and sample nested within taxon as a random effect. Since some samples exhibited no incorrect match, we added 0.001 to the proportions before logit transformation. We fitted three models, each at a different step in dataset assembly: one for raw reads, another for all assembled loci for each sample and another only including loci in the final dataset. Fitting of linear mixed models was performed as described above, and all pairs of samples sharing at least 100 loci in the final dataset were included, totaling 57 pairs of libraries. The same samples were included in all three models (Table 1).

Contamination between samples within a species would be harder to perceive and more problematic in most contexts. Even though we cannot detect it directly, it is possible to test whether the effects of this form of contamination cause a significant bias. To identify samples that likely have a significant amount of contamination from close relatives, we compared the overall genetic distance of SNP sites between the two libraries (gDNA and MDA) produced for each sample. We used as a distance metric the average number of pairwise differences between the alleles of two libraries at SNP sites, ignoring Ns and gaps. For each gDNA library, we also obtained the distance between this library and gDNA libraries from other individuals from the species, if at least 35 loci were shared. The lower filter for number of loci used here was necessary to include all libraries in the comparison and, as shown in the results, libraries with very few loci shared with other samples were those with most problems. With little to no contamination, MDA libraries are expected to be very similar to the gDNA libraries from the same sample, and more similar than gDNA libraries produced from closely related individuals. Therefore, we identified problematic samples as those in which their gDNA and MDA libraries have higher genetic distances between themselves than the gDNA library has to its most similar conspecific gDNA library. Considering that our mixed models for contamination between taxa indicated that samples with higher more loci might be less affected by contamination, we also observed whether samples identified as problematic using the procedures above are among those with fewer loci in the final dataset.

## Overall effect of biases caused by MDA

In addition to understanding the effects of MDA on specific parameters, such as heterozygosity, GC bias, and amount of contaminants, we tested whether the overall biases and errors caused by MDA have the potential to overcome the natural genetic variability of the populations studied here and impact a population-level study. To understand if this is the case, we first excluded the nine MDA libraries with the smallest number of loci in the final dataset, since our results showed that samples with significant levels of contamination are among those with large amounts of missing data. Next, we have generated, for each taxon, a matrix of average pairwise genetic differences in SNP sites. Since in a few cases two libraries for the same taxon did not have any loci in common, a pairwise distance could not be calculated and both libraries were removed from this analysis, resulting in a total of 48 samples with two libraries each across five taxa. We then modeled the relationship between this distance matrix as a response and several predictors (MDA, population and number of loci) using a MDMR. The aim of this analysis is to test whether biases in MDA lead to spurious clustering of libraries produced from different individuals. We further checked graphically whether libraries from a single sample clustered together by producing neighbor-joining trees based on the matrix of genetic distances.

As a proof of concept that MDA produces high-quality data for studies of phylogeography, we have also performed population genetics analyses. Following our own recommendations of imposing stringent filters for sample and locus inclusion in the final dataset, we first filtered both samples and loci to reduce the amount of missing data. In the ipyrad pipeline, this is done by removing specific samples and setting a minimum number of samples per locus, and we built an interactive visualization tool using the R package shiny (*Chang et al., 2018*) (Matrix Condenser, available at https://bmedeiros.shinyapps.io/matrix_condenser and https://github.com/brunoasm/ matrix_condenser) to identify an optimal combination of sample and locus filtering to reduce missing data. We aimed to retain as many samples and loci as possible for each taxon with no more than 50% missing data per sample, and only removed samples based on the amount of missing loci, not in their identity or library preparation method.

For each taxon, we generated a new branch in the ipyrad pipeline using the parameters chosen in matrix condenser. We then exported new output files containing unlinked SNPS and converted these files from phylip to genpop format using a custom python script (Data S3). We did a principal component analysis of the genetic variation and *k*-means clustering (*Jombart, Devillard & Balloux, 2010*) in the R package adegenet (*Jombart, 2008*; *Jombart & Ahmed, 2011*), visualizing whether clusters obtained were compatible with geography (i.e., populations) or method of library preparation. Here, we are mostly concerned with whether MDA results in spurious clusters, and detailed population genetic analyses with full description of the populations involved and a larger sampling throughout the range of the species will be published elsewhere.

For *C. impar*, we further tested the effects of MDA in the estimation of $F_{ST}$ for studying isolation by distance. Since MDA may potentially bias heteozygosity, it could also have

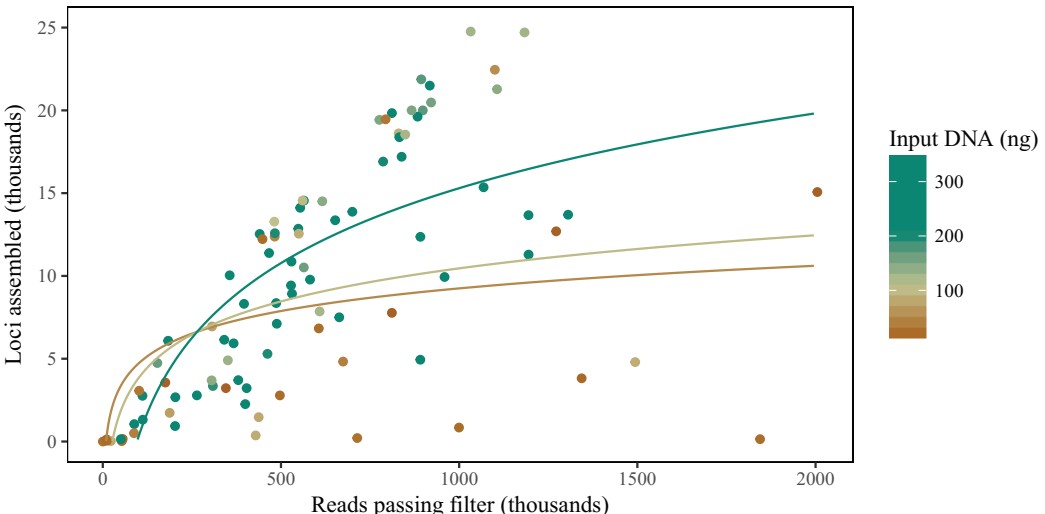

**Figure 2 Small amounts of input genomic DNA in the MDA reaction reduce the number of loci assembled for a given number of raw reads.** Here, we included all MDA libraries, whether or not there was a gDNA library prepared from the same sample. Colors used for both lines and points follow the same scale. The three lines represent model predictions for the relationship between number of reads and number of loci for input DNA amounts of 50 ng (brown), 100 ng (beige), and 300 ng (green).

effects on $F_{ST}$. We chose to focus only on *C. impar* for this analysis because other taxa either resulted in too few pairwise population comparisons (*Andranthobius*, *M. bondari*, and *M. ypsilon*, Fig. 1) or are more complex datasets containing cryptic species (*Anchylorhynchus*), which require analyses beyond the scope of this paper. For *C. impar*, we estimated pairwise $F_{ST}$ between populations and estimated the correlation of linearized $F_{ST}$ with pairwise geographical distance. Significance of this correlation was tested by using a Mantel test in R package vegan (*Oksanen et al., 2018*). We repeated the analysis including only MDA samples or only gDNA libraries, and observed whether correlation differed between library preparation methods. $F_{ST}$ was estimated using the unbiased estimator of *Weir & Hill (2002)* combined across loci according to *Bhatia et al. (2013)*, implemented in the R package BEDASSLE (*Bradburd, Ralph & Coop, 2013*). Great circle distances were calculated from geographical coordinates using the R package fossil (*Vavrek, 2011*). All R scripts used to produce statistical results and figures in this structure provided as R markdown notebooks in Article S2 (MDMR and mixed models) and Article S3 (population genetics analyses). All Python and bash scripts are available in Data S2 (MDMR and mixed models) and Data S3 (population genetics analyses).

## RESULTS

### Effect of starting DNA amount on the number of loci assembled for MDA libraries

The amount of input DNA in the MDA reaction has a significant effect on the number of assembled loci for a library (Table 1; Fig. 2). When small amounts of input DNA are used, the average number of loci obtained decreases, but this effect is highly

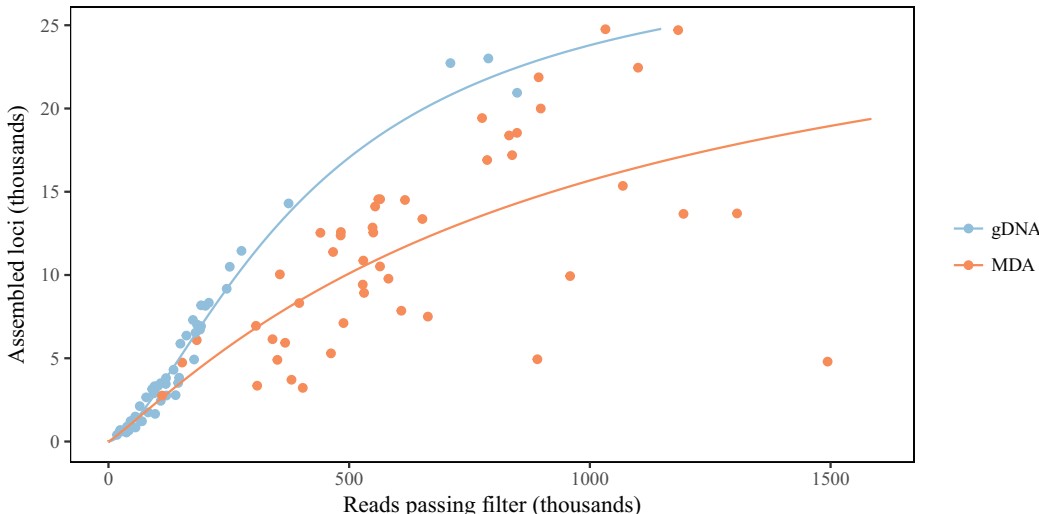

**Figure 3 MDA reduces the number of loci assembled for a given number of reads.** Colored lines represent predictions from the mixed model for each type of library, averaging random effects.

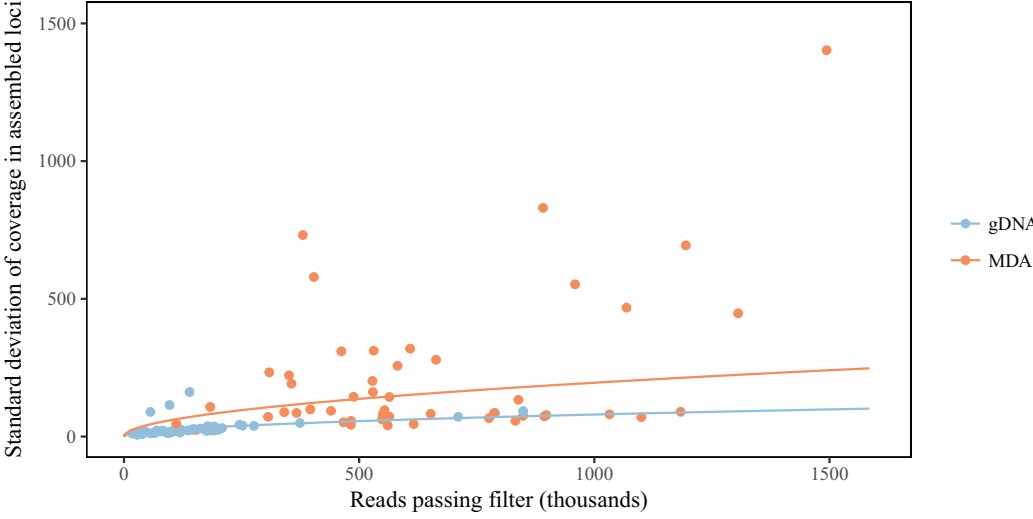

**Figure 4 MDA increases variation in read depth across assembled loci.** Colored lines represent predictions from the mixed model for each type of library, averaging random effects.

variable: some MDA libraries prepared with very little DNA are comparable to gDNA libraries, while others result in very few loci even with high sequence coverage.

## Comparisons between MDA and gDNA libraries for loci assembled, GC content, variation in coverage, and heterozygosity

When compared to their gDNA counterparts, MDA reduces the average number of loci obtained for a library, when controlling for the number of reads, and also seems to increase the variation of the outcome (Table 1; Fig. 3). Even though the coefficient for

**Table 2 Results of MDMR model fitting for dissimilarity in the set of sequenced loci for each taxon.**

| Taxon | Predictor | Pseudo $R^2$ | $p$-Value |
|---|---|---|---|
| *Anchylorhynchus* 82 libraries | Omnibus effect | **0.2573** | **<0.002** |
| | MDA | **0.0148** | **0.034** |
| | Pool | **0.0547** | **0.006** |
| | Population | **0.1255** | **<0.002** |
| | Loci in final dataset | **0.0214** | **<0.002** |
| *Andranthobius* 27 libraries | Omnibus effect | **0.4095** | **<0.002** |
| | MDA | **0.0478** | **0.028** |
| | Pool | **0.105** | **0.002** |
| | Population | 0.0701 | 0.112 |
| | Loci in final dataset | **0.133** | **<0.002** |
| *Celetes impar* 43 libraries | Omnibus effect | **0.2911** | **<0.002** |
| | MDA | 0.0239 | 0.344 |
| | Pool | 0.0239 | 0.346 |
| | Population | **0.1214** | **0.020** |
| | Loci in final dataset | **0.0794** | **<0.002** |
| *Microstrates bondari* 33 libraries | Omnibus effect | **0.2704** | **<0.002** |
| | MDA | 0.0378 | 0.06 |
| | Pool | 0.062 | 0.192 |
| | Population | 0.0664 | 0.054 |
| | Loci in final dataset | **0.0696** | **<0.002** |
| *Microstrates ypsilon* 31 libraries | Omnibus effect | **0.2975** | **<0.002** |
| | MDA | 0.0307 | 0.608 |
| | Pool | 0.0643 | 0.258 |
| | Population | 0.0922 | 0.434 |
| | Loci in final dataset | **0.0527** | **<0.002** |

**Notes:**
All paired and non-paired gDNA and MDA libraries are included. Omnibus effect is the combined effect of all predictors. Pseudo $R^2$ values are conceptually similar to $R^2$ values in a typical linear regression (*McArtor, 2017*). Significant $p$-values are highlighted in bold. Samples in the same pool have undergone size selection and PCR together.

MDA alone seems to suggest the opposite effect, the interaction term with number of reads results in a predicted decrease of at least 29% in the number of loci assembled for number of reads above 200,000 (Fig. 3). While reducing the number of loci being assembled, MDA also increases the variation in coverage between loci (Table 1; Fig. 4). There is evidence that this smaller number of assembled loci is a biased set. In three of the five taxa we found significant effects of MDA on the proportion of loci shared between libraries, but the variation explained by MDA was typically half of that explained by samples being pooled together for size selection as measured by pseudo-$R^2$. Population identity and overall number of loci obtained for a sample also had generally greater effects (Table 2).

For both MDA libraries and gDNA libraries, heterozygosity increases with the number of loci, with the effect being more pronounced in case of MDA (Table 1; Fig. 5A). When the average coverage of assembled loci is used as predictor,

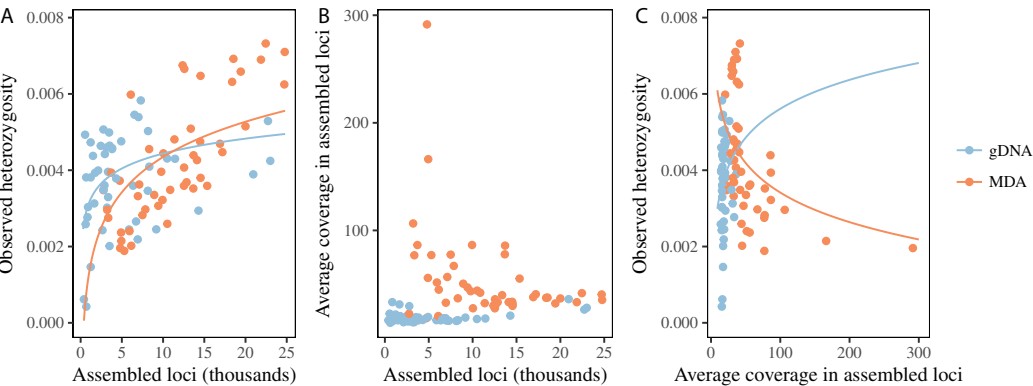

**Figure 5 Heterozygosity depends on number of loci and average coverage, and MDA affects these relationships.** Colored lines represent predictions from the mixed model for each type of library, averaging random effects. (A) Heterozygosity increases with number of loci assembled, and the relationship is steeper for MDA libraries. (B) For gDNA libraries average coverage slightly increases with number of loci. For MDA libraries, samples with unusually low number of assembled loci have unusually large average coverage per locus. (C) For gDNA libraries, heterozygosity increases with average coverage. The opposite is observed for MDA libraries. The depression in heterozygosity is more severe in samples with very high average coverage.

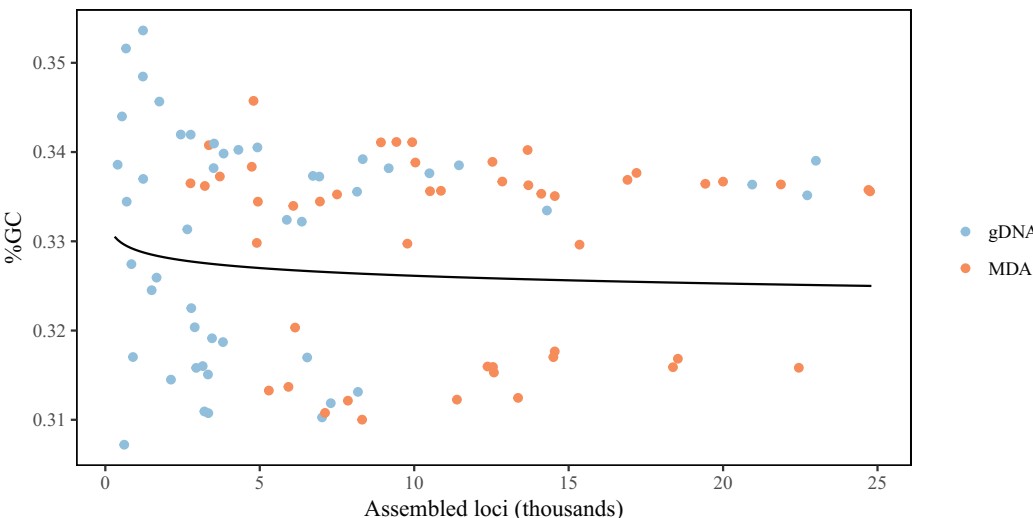

**Figure 6 MDA does not affect the GC content of loci obtained using ddRAD.** Lines represent predictions from the mixed model averaging random effects, and colors represent the kind of library.

we observe a positive relationship in the case of gDNA libraries but negative in the case of MDA libraries (Table 1; Fig. 5C). For gDNA libraries, the number of assembled loci and average coverage are positively correlated (Fig. 5B), but for MDA libraries this relationship is inverted because libraries assembling an unusually low number of loci also have unusually large average coverage per locus (Fig. 5B). Together, we find that average coverage and heterozygosity generally increase with the number of reads for both gDNA and MDA libraries, but in a few cases MDA libraries result simultaneously in a low number of loci and low heterozygosity. The GC content of

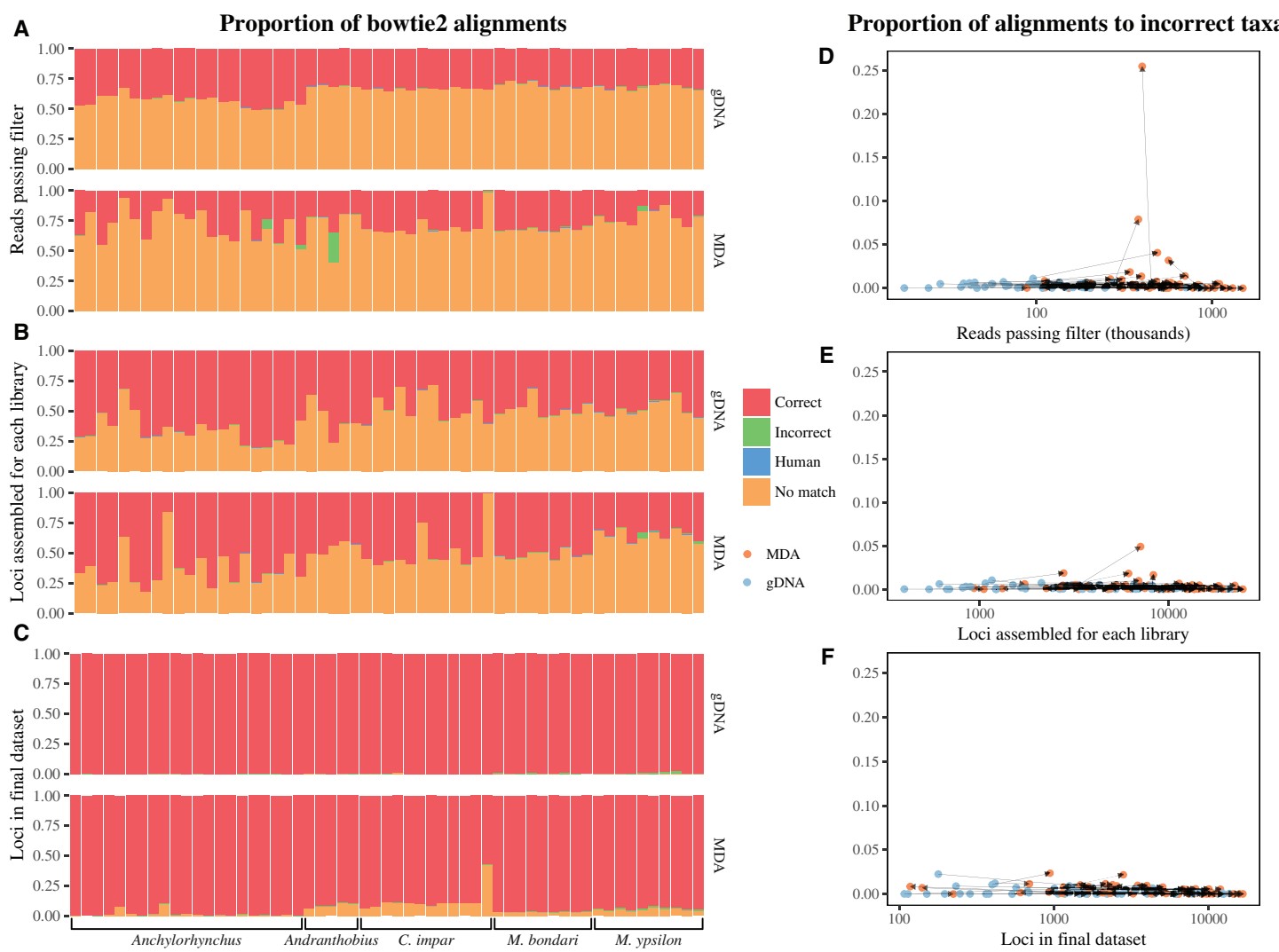

**Figure 7 Effect of MDA on non-templated amplification on reads (A, D), assembled loci (B, E) and loci in the final dataset (C, F).** In all graphs, all libraries are included. Only reads and loci are excluded due to filters imposed during successive bioinformatics steps (Fig. 1). For most samples, there is no detectable effect of MDA. In the few cases with large proportion of raw reads matching to the incorrect taxon, this is drastically reduced at each step in the dataset assembly. (A–C): Each bar represents a sample, with colors representing the proportion of reads/loci aligning to each source in the reference dataset. Correct: sequences that match to the correct taxon, Incorrect: sequences that match another weevil species, Human: sequences that match the human genome, No match: sequences that do not match any sequence in the reference dataset. Taxa are indicated in the bottom. (D–F): Proportion of reads/loci aligning to incorrect taxa (red in the bar graphs) against number of reads/loci sequenced, with scale of Y axis constant. Arrows connect corresponding MDA and gDNA libraries.

assembled loci seems to decrease slightly with the number of loci, but it is not affected by MDA (Table 1; Fig. 6).

## Non-templated amplification

Contamination by human DNA is negligible in both kinds of libraries, with the main source of identifiable non-template sequences matching the other beetle taxa included in this study (Fig. 7). In general, most MDA libraries show a larger share of reads and loci not matching the reference dataset than their gDNA counterparts. Given that the

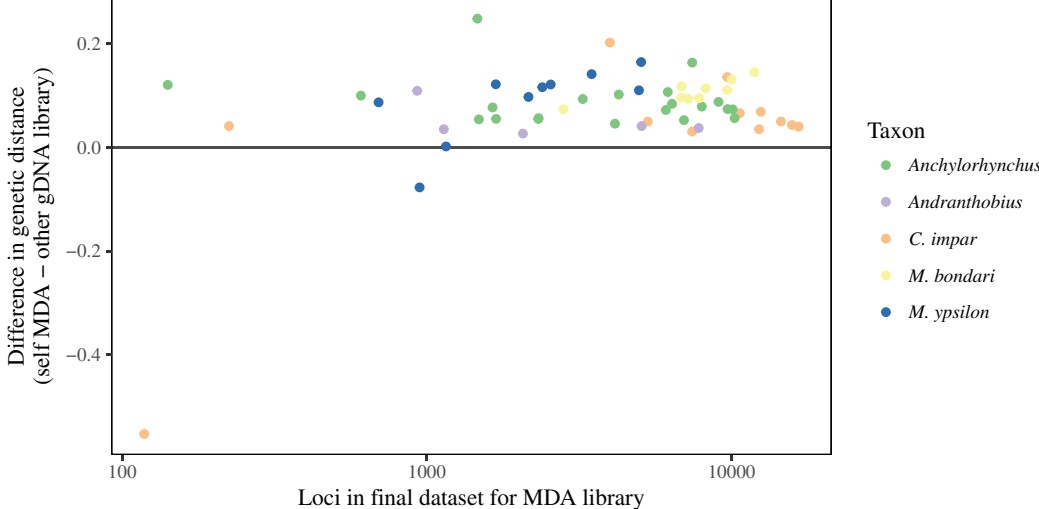

**Figure 8 Difference between genetic distances to MDA and gDNA libraries.** For each gDNA library, values in the *Y*-axis are the difference between its genetic distance to its corresponding MDA library and the distance between this gDNA library and its most similar gDNA library prepared from a different sample. Values below 0 (horizontal line) indicate cases in which MDA caused enough bias as to make the genetic distance within a single individual larger than that between individuals. The three cases with evidence for such bias are among the samples with fewer loci shared in the final dataset.

reference set was made from loci in gDNA libraries, this is not surprising. However, the proportion of unidentifiable loci is small in the final dataset (Fig. 7). When compared to their corresponding gDNA libraries, the increase in the proportion of reads/loci matching to sequences of an incorrect taxon for MDA libraries is generally small, with only a few samples showing a very large share of incorrect matches in reads, which diminishes in the final dataset (Fig. 7). Linear mixed models did not find a significant effect of MDA in the proportion of reads matching to incorrect taxa (Table 1), but a higher number of reads is significantly associated with a smaller proportion of incorrect matches. In the case of assembled loci, we did not find a significant effect of either MDA or number of loci (Table 1). In the final dataset, we found that MDA slightly increases the proportion of loci matching to incorrect taxa, and this proportion decreases with a larger number of loci (Table 1). Even though the effect is significant in the final dataset, its magnitude is small and the overall proportion of incorrect matches is reduced at each bioinformatics step. The most extreme outlier in raw reads is represented by an MDA library that had an excess of 25% of reads matching to an incorrect taxon when compared to its gDNA counterpart (Fig. 7). The highest increase in incorrect matches for MDA samples in loci in the final dataset, on the other hand, was 1.3%, and the average increase was 0.1% (Fig. 7).

Only three libraries exhibited potential bias caused by MDA when comparing the genetic distance between libraries, with distances between gDNA libraries prepared from different individuals being equal to or smaller than the gDNA and MDA libraries prepared from the same sample (Fig. 8). All of these MDA libraries are among the nine (15%) with the smallest number of loci shared in the final dataset, suggesting that

**Table 3 Results of MDMR model fitting for pairwise genetic distances in each taxon.**

| Taxon | Predictor | Pseudo $R^2$ | *p*-Value |
|---|---|---|---|
| *Anchylorhynchus* 32 libraries | Omnibus effect | **0.9025** | **<0.002** |
| | MDA | 0.0089 | 0.126 |
| | Population | **0.8503** | **<0.002** |
| | Loci in final dataset | 0.0043 | 0.358 |
| *Andranthobius* six libraries | Omnibus effect | 0.5426 | 0.664 |
| | MDA | 0.0653 | 0.890 |
| | Population | 0.3834 | 0.228 |
| | Loci in final dataset | 0.0355 | 0.956 |
| *Celetes impar* 20 libraries | Omnibus effect | **0.6500** | **<0.002** |
| | MDA | 0.0132 | 0.848 |
| | Population | **0.6059** | **<0.002** |
| | Loci in final dataset | 0.0302 | 0.332 |
| *Microstrates bondari* 18 libraries | Omnibus effect | **0.4850** | **<0.002** |
| | MDA | 0.0465 | 0.334 |
| | Population | **0.3954** | **<0.002** |
| | Loci in final dataset | 0.0644 | 0.114 |
| *Microstrates ypsilon* 14 libraries | Omnibus effect | **0.5620** | **<0.002** |
| | MDA | 0.0328 | 0.710 |
| | Population | **0.4089** | **<0.002** |
| | Loci in final dataset | 0.0879 | 0.048 |

**Notes:**
Only paired gDNA and MDA libraries sharing loci with all samples in the final dataset are included. Omnibus effect is the combined effect of all predictors. Pseudo $R^2$ values are conceptually similar to $R^2$ values in a typical linear regression (*McArtor, 2017*). Significant *p*-values are highlighted in bold.

samples resulting in unusually low number of loci are more likely to have undergone significant non-templated amplification. For all further analyses, to avoid the inclusion of samples with significant amount of non-template loci in the final dataset, we removed these nine MDA libraries.

## Overall effect of biases caused by MDA

None of the five taxa exhibited an effect of MDA on clustering by sequence similarity, and in all cases the pseudo-$R^2$ associated with population assignment was much larger than that associated with MDA (Table 3). Moreover, in neighbor-joining dendrograms all pairs of libraries prepared from the same sample correctly cluster with each other (Fig. S1). In population genetics analyses including both MDA and gDNA samples filtered to reduce missing data (Table S2), we observe genetic clusters that correspond to cryptic species or to geography (other taxa), not to library preparation method (Fig. 9). For *Anchylorhynchus*, clusters correspond to cryptic species. For *Andranthobius*, there is little genetic structure between the three localities sampled and all samples are recovered as part of the same cluster. For *C. impar*, all local populations are contained in one of the genetic clusters. For *M. ypsilon*, only two clusters are found, and one of the populations has individuals in both of them. This population is also geographically

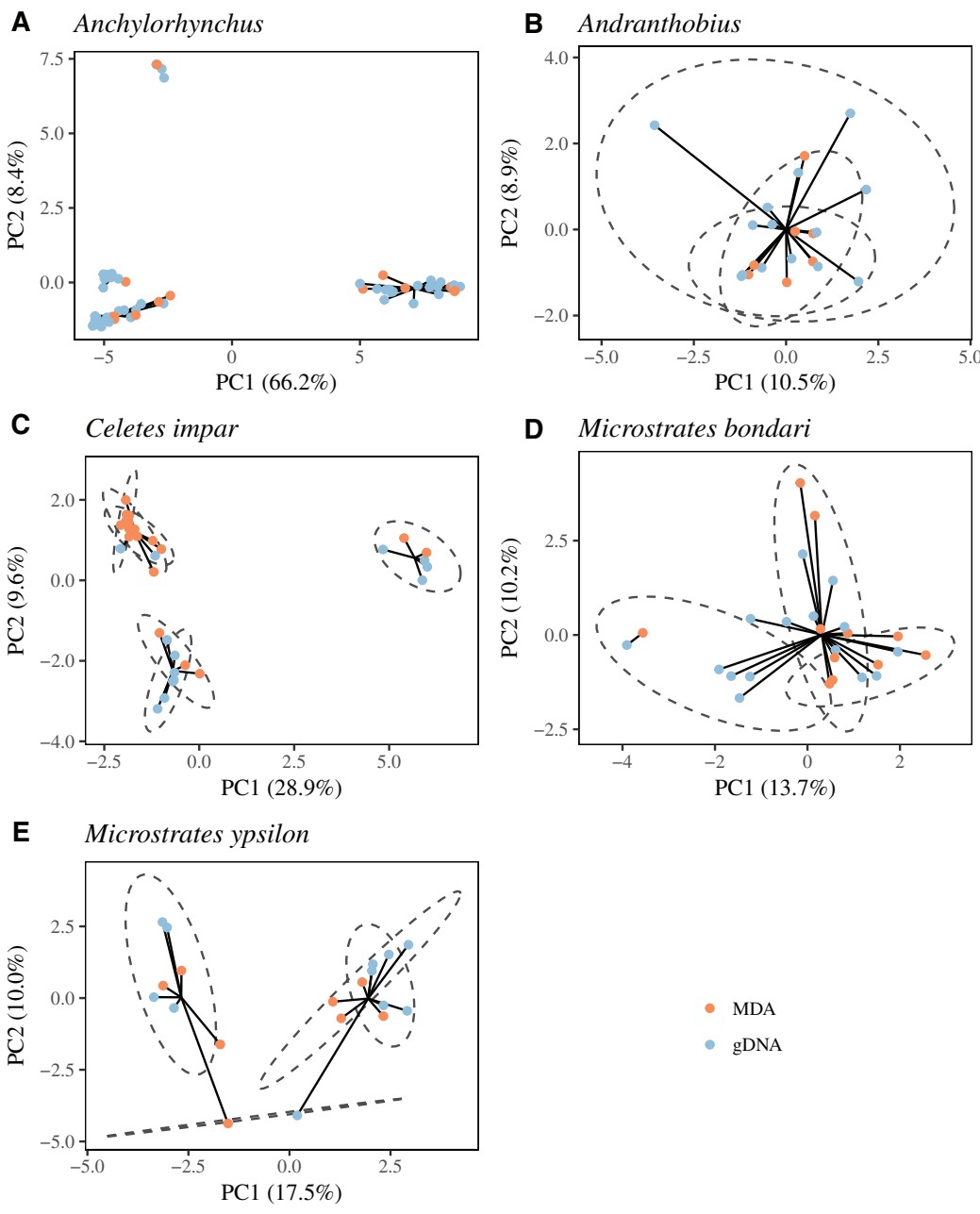

**Figure 9 Principal component analyses and *k*-means clustering of weevil populations.** (A) *Anchylorhynchus*, (B) *Andranthobius*, (C) *Celetes impar*, (D) *Microstrates bondari* and (E) *Microstrates ypsilon*.

positioned between the other two. For *M. bondari*, two clusters are found. One includes most individuals while the other includes one MDA and one gDNA sample. The population divided between two genetic clusters is also the most geographically isolated one. In summary, even though patterns differ between taxa, in no case we observe clustering by library preparation method. For *C. impar*, patterns of isolation by distance using only MDA libraries, only gDNA libraries or both together are

**Table 4 Mantel tests for isolation by distance in *Celetes impar*.**

| Dataset | Number of populations | Correlation | *p*-Value |
|---|---|---|---|
| All samples | 6 | 0.4802 | 0.025 |
| gDNA | 4 | 0.4773 | 0.0833 |
| MDA | 6 | 0.5076 | 0.025 |

**Note:**
See Table S2 for the number of gDNA and MDA libraries included in each population.

essentially the same (Table 4; Fig. S2). The correlation between geographical distance and $F_{ST}$ is very similar in these three cases, even though not significant at the 0.05 level for gDNA alone since in the filtered dataset two populations did not have any gDNA sample (Table 4).

## DISCUSSION

The comparison between libraries produced from the same sample with and without MDA reveals that MDA can result in high-quality libraries from samples with little DNA yield, but some caveats should be observed. First, there is generally a decrease in the number of assembled loci for a given number of reads (Fig. 3). This contrasts with the results of *Blair, Campbell & Yoder (2015)*, who did not find any difference in the number of loci recovered from MDA and direct libraries. This difference might be explained, at least in part, by the smaller amount of input DNA used in some samples here, since we found a significant effect of amount of input genomic DNA in an MDA reaction on the number of loci, after controlling for number of reads obtained (Fig. 2). It is likely that, when low amounts of input DNA are used, stochastic effects during genome amplification may result in uneven representation of different genomic regions, decreasing the coverage of some loci below the defined threshold (*Sabina & Leamon, 2015*). Deterministic effects could also cause a reduction in the number of assembled loci, since we found evidence for gDNA and MDA libraries being associated with different sets of loci in at least some taxa. This effect, however, seems to be small when compared to the effect caused by natural genetic variation between populations and of pooling samples for size selection and PCR (Table 2). We have not found biased amplification of regions with different GC content, as reported for MDA in previous studies (*Sabina & Leamon, 2015*). It is likely that the main driver of GC biases in RAD libraries is PCR amplification (*DaCosta & Sorenson, 2014*), not MDA. One deterministic effect that has been reported and is consistent with our results is the differential amplification of loci by size. *Shortt et al. (2017)* found that loci recovered in MDA-amplified samples are consistently shorter than loci in non-amplified samples, even with the same size selection protocol. In the absence of reference genomes, we cannot evaluate fragment size here, but our results are consistent with a mixture of deterministic and stochastic effects resulting in the smaller set of loci for MDA samples.

We have found evidence for the presence of raw reads resulting from non-templated amplification in a few samples, but across all samples it does not seem to be significant. Most of the reads that do not align to the correct species do not have a clear origin, but in a
few cases they seem to result from contamination from other samples (Fig. 7). MDA, therefore, might add another step in which cross-contamination can occur. Even though there is some evidence for contamination in the sequenced reads, filters imposed by bioinformatics steps seem to eliminate or reduce the problem. Even in the few cases in which there is a large number of reads resulting from contamination from other species, these seem to be largely eliminated from the final dataset, probably because the randomness in sources of contamination for the different samples results in no shared loci derived from contaminants. It is important to highlight that we imposed non-stringent filters for minimum coverage at a locus (seven) and of number of samples sharing a locus (four) for the tests of contamination. It is likely that more stringent criteria will have an even greater effect in eliminating potential contaminants.

We have also observed some evidence for contamination between closely related samples (Fig. 8), with MDA resulting in a few libraries that are genetically dissimilar to their corresponding gDNA library. These libraries all exhibit a low number of loci in the final dataset, when compared to other libraries for the same taxon. At least for the cases observed here, filtering libraries sharing only a few loci with others in the final dataset was sufficient to eliminate problematic samples. Since exclusion of samples with high amounts of missing data is already a requirement for some commonly used methods in population-level studies, such as fastSTRUCTURE (*Raj, Stephens & Pritchard, 2014*), ADMIXTURE (*Alexander, Novembre & Lange, 2009*), and DAPC (*Jombart, Devillard & Balloux, 2010*), it is likely that datasets prepared for population genetic analyses based on whole-genome patterns of variation would not contain significant biases.

Even though MDA generates reliable data for population genetics, and has been successfully used to study the number of queens in ant colonies (*Boyle et al., 2018*), it might be problematic for other applications of ddRAD. For phylogenetic studies, missing data due to indels and mutations in restriction sites are expected to be common, especially among more distantly related species (*Rubin, Ree & Moreau, 2012*). It has even been suggested that missing data should not be excluded from phylogenetic datasets produced from RADseq (*Eaton et al., 2016*; *Tripp et al., 2017*) and that patterns of missing loci can be used as phylogenetic characters (*DaCosta & Sorenson, 2016*). This contrasts with our recommendation of excluding samples with high levels of missing data, and it might be hard to distinguish library preparation artifacts from biological variation in the number of loci. Another potential use of ddRAD in which MDA might be inadequate are genome scans for signatures of selection. Even a very small level of contamination might result in false positive outlier loci, adding to the potential problems with using RAD in studies of adaptation (*Catchen et al., 2017*; *Lowry et al., 2017a*, *2017b*; *McKinney et al., 2017*).

Multiple displacement amplification seems to have some effect on genotyping, since the effect of number of loci and average coverage per locus is different in MDA and gDNA libraries (Fig. 5). MDA libraries with small number of assembled loci are associated with large average coverage per locus and low observed heterozygosity. This indicates that, in some samples, MDA over-amplified one of the alleles for some loci, resulting in few

loci with large coverage and apparent homozygosity. As in the case of contamination, samples yielding fewer loci in the final dataset seem to be more strongly affected. Using a fairly large amount of template DNA in the MDA reactions, *Blair, Campbell & Yoder (2015)* did not observe loss of heterozygosity in MDA libraries compared to gDNA libraries.

Even though MDA does introduce small errors, as we observed in heterozygosity, and a small level of contamination in a few samples, these seem to be insignificant when compared to the natural genetic variation of the species that we studied, and does not seem to introduce systematic bias. We did not find a significant effect of MDA on the pairwise genetic distances between libraries (Table 3), and neighbor-joining trees based on the pairwise genetic distances clearly recovers individuals and populations (Fig. S1). Further, in tests with datasets filtered to exclude samples and loci with large amounts of missing data we did not find any evidence for *k*-means clustering or $F_{ST}$ being biased when MDA and gDNA libraries are used in the same analysis (Fig. 9; Table 4; Fig. S2).

An important limitation of our study is the fact that sampling and sequencing was not initially designed to test the effects of MDA, but rather to study the phylogeography of the species included here. A major consequence is the uneven number of reads obtained between gDNA and corresponding MDA libraries, since MDA libraries were sometimes produced specifically for samples for which more coverage was desired. This difference in number of reads passing filter should be not be understood as a result of MDA, since MDA and gDNA samples were generally sequenced separately. Rather, we considered it as an independent variable that needs to be accounted for while modeling. Therefore, we addressed the problem of variation in number of reads per sample by removing libraries with very few assembled loci from the dataset and by explicitly including the number of reads or assembled loci as predictors in our models. Another important caveat is that, since variability in the amount of input gDNA in the MDA reaction came from the natural variation in samples, it is possible that the effect of decreasing DNA amounts that we observed is not simply a consequence of DNA quantity. It is reasonable to assume that, for a given species, samples that yielded less DNA are also likely to have lower-quality DNA, and therefore the effect might be largely driven by DNA quality, not quantity. It is noteworthy that a relationship between sample degradation and a lower number of sequenced loci has been found for ddRAD before, even in the absence of whole-genome amplification (*Graham et al., 2015*). Considering that previous studies obtained even genomic coverage with input amounts of high-quality DNA for the MDA reaction on the order of one ng (*Ng, Roberts & Coleman, 2005*), it is likely that DNA quality also plays a role here and care should be taken when using MDA for sheared samples as some museum specimens. The same correlation between DNA quantity and quality is probably encountered in other studies in which MDA might be useful. As a consequence, we can expect that, in general, MDA can be used to rescue a sample with little DNA yield that would otherwise be unsuitable for ddRAD, with a possibility that the number of loci recovered will be lower than expected for a typical sample.

## CONCLUSION

We characterized the effects of MDA on ddRAD libraries by making comparisons in conditions experienced in studies in which MDA is most needed. Whole-genome amplification by MDA is an option for increasing the starting amount of DNA for samples that would usually not be suitable for ddRAD. The main problem resulting from MDA seems to be a reduction in the number of loci obtained for a given number of reads, which can increase the sequencing costs in addition to the cost of MDA procedure itself. Other artifacts, such as contamination and genotyping bias, are not found consistently across samples and their effects are negligible for the purposes of population genetic studies after samples with an unusually low number of sequenced loci are filtered. In general, our sampling resulted in data generated under conditions similar to those encountered in studies sequencing a large number of samples from organisms with little DNA yield, and the robustness of MDA that we find here indicates that it is useful, even in non-ideal conditions, in the preparation of ddRAD libraries. When proper care is taken in the assembly of datasets, MDA coupled with ddRAD offers the possibility of generating high-quality libraries for genomic studies in small organisms.

## ACKNOWLEDGEMENTS

We are thankful to Sergio A. Vanin and Sonia Casari, curators of the Museum of Zoology of the University of São Paulo, and the museum staff for the assistance with specimen handling and loan. We also thank the staff of the Bauer core facility at Harvard for training and assistance with instruments. The computations in this paper were run on the Odyssey cluster supported by the FAS Division of Science, Research Computing Group at Harvard University. This manuscript was greatly improved with comments from John D. Boyle, Tauana J. Cunha, three anonymous reviewers and the handling editor.

### Funding

This work was supported by the Harvard University William F. Milton Fund and the Harvard University Department of Organismic and Evolutionary Biology Graduate Research Fund. Fieldwork was supported by the Museum of Comparative Zoology Putnam Expedition Grant and David Rockefeller Center for Latin American Studies Research Travel Grant. Bruno de Medeiros received a Jorge Paulo Lemann Fellowship for Research in Brazil. The funders had no role in study design, data collection and analysis, decision to publish, or preparation of the manuscript.

### Grant Disclosures

The following grant information was disclosed by the authors:
Harvard University William F. Milton Fund and the Harvard University Department of Organismic and Evolutionary Biology Graduate Research Fund.
Museum of Comparative Zoology Putnam Expedition Grant and David Rockefeller

Center for Latin American Studies Research Travel Grant.

Bruno de Medeiros received a Jorge Paulo Lemann Fellowship for Research in Brazil.

## Competing Interests

The authors declare that they have no competing interests.

## Author Contributions

- Bruno A. S. de Medeiros conceived and designed the experiments, performed the experiments, analyzed the data, contributed reagents/materials/analysis tools, prepared figures and/or tables, authored or reviewed drafts of the paper, approved the final draft.
- Brian D. Farrell contributed reagents/materials/analysis tools, authored or reviewed drafts of the paper, approved the final draft.

## Field Study Permissions

The following information was supplied relating to field study approvals (i.e., approving body and any reference numbers):

All samples studied here were collected with permits from the Instituto Chico Mendes de Preservação da Biodiversidade (permit number 39704-7) and deposited in the Museum of Zoology of the University of São Paulo.

## DNA Deposition

The following information was supplied regarding the deposition of DNA sequences:

Sequences assembled by ipyrad are provided as *.loci files in Data S2.

## Data Availability

Codes are deposited in Bruno A.S. de Medeiros's Github repository (http://www. github.com/brunoasm) as well as in the Supplemental Files.

## Supplemental Information

Supplemental information for this article can be found online at http://dx.doi.org/ 10.7717/peerj.5089#supplemental-information.

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
