# Peer review of "Whole-genome amplification in double-digest RADseq results in adequate libraries but fewer sequenced loci"

_PeerJ, doi:10.7717/peerj.5089_

## Round 0.1 · original submission · Minor Revisions

Your manuscript has been seen by three qualified reviewers (the report provided as "Reviewer #2" actually contains comments from two people from the same lab). Based on their detailed assessment and my own, I feel this work is well suited for publication in PeerJ pending a series of minor changes.

Reviewer 1 ·

Basic reporting

good. most of my comments are in the general section. Minor errors or requested clarifications are noted there.

Experimental design

I would like to see a little bit more analysis done, as noted below, but this should not be too big a problem. I hope that the questions about clone fragment filtering can be dealt with.

Validity of the findings

Good. Some comments on the discussion, phrasing of conclusion at end of abstract that I thought was a bit overly strong given what was presented.

Additional comments

Overall, the paper seems reasonable and I would say that results are broadly in agreement with my own experience.

Was there no process carried out in the analysis for the removal of clones? If so, I missed it. This is fairly standard, I think. This is going to be particularly important for low DNA starting material in the context of this study.

A main point is that the number of loci recovered with MDA is not expected to be lower based on Blair et al, but in Shortt et al (which you cited elsewhere) there was a clear redistribution of lengths of the loci recovered with MDA towards shorter length, which directly predicts that the number of loci that would achieve some depth of sequencing would also shift (as you found). This is worth commenting on, and also potentially worth describing the shift in length of loci recovered in your data (including how many sequences are wasted on loci that are too short to included in the analysis). This point is in addition to, and possibly much larger effect than the stochastic shift (cited Sabina and Leamon).

The last sentence of the abstract (“Overall, we find 37…) would be better with some kind of qualifier about the relative development, study design, and utilization costs of “target enrichment” in terms of method development and producing less data. This is especially because the current paper does not include a direct comparison of this other method. I don’t disagree with most of the results, but I am not convinced that I would or should even consider switching to target enrichment.

I am not too fond of the numbered analyses format. Is that by author choice or PeerJ?

On line 94, are the authors implying that they can make libraries regularly off of a few ng of DNA without amplification? I am quite impressed if so. The Andrews et al paper cited mentions 50-100 ng of DNA “can often be implemented”, which seems more reasonable in my own experience.

The discussion on lines 389-397 about potential problems with phylogenetics was not particularly clear to me. I’m not sure that “locus dropout” is the most straightforward term for differences in locus existences between species, as it is more suggestive of low frequency variants in a population. The switch from missing loci in samples to excluding data and that that is a problem with “less redundant” sampling in phylogenetic studies was particularly confusing. What point is being made? It seems clear that there will be fewer loci that can be compared between species than within populations, but still a lot. If a sample didn’t amplify well enough to provide reliable information, it isn’t that useful data is being left out, but rather limited and potentially misleading data. At any rate, please clarify the point.

In the discussion on biased recovery of different loci, (line 402 on), this was mentioned earlier, but I don’t see why the authors don’t look at the bias due to MDA in recovery, rather than speculating that it is a smaller difference than size selection. Further, the fact that thousands of loci align between datasets may trivially mean that they are overlapping, but it does not say clearly how much overlap there is as opposed to wasted sequencing due to non-overlap. This can be assessed directly, and should be.

Line 95: would be better without redundant “useful…use”. Suggest “MDA may enable the use of…”

Line 168: I don’t know what would make 7 “the minimal threshold”, which makes it sound like a necessary boundary, although it is probably a reasonable lower bound. Is this due to the program used to call, or is it based on some accuracy criterion? Please mention in either case.

Line 175: The minimum of 4 samples for a locus seems somewhat random and unmotivated as written. Perhaps you could justify it? I would think that a reasonable criterion would depend greatly on the number of samples per species, and if it is 4 across all the species, it would seem to arbitrarily depend on how close the species are. Please clarify how and why this cutoff was (and should be, for future studies) chosen.

Line 183: “we have counted” makes it sound like you did it once, maybe not related to this paper. Better to use direct past tense, as in, “To test, …we counted…” The recommended change is also the same format as e.g., Line 197, making it consistent.

Line 275: fix “are have” and plural consistency, use of “higher”. For example, “MDA libraries have greater genetic distances among themselves than the gDNA library has…”

Line 277: change “higher number of” to “more”

Line 278: change “have also observed” to “also observed” for tense consistency and readability.

Line 290: change to “loci in common”

Line 311: the first part of the sentence just repeats what was said in the previous two sentences.

On line 311-315, this is a question where the analysis of clonality among the reads would impact interpretation of the results.

On line 331, please go through the manuscript and eradicate the use of “higher number”. The correct term here is “larger numbers” or just “more”, where appropriate. Also, I believe the intended plural usage would make this “these proportions decrease with larger numbers of loci”.

On line 332, “in fact” is a colloquialism that doesn’t belong in a scientific paper.

Change on line 394 to “…should not be…”

On line 440, was this meant to be “on the order of 1 ng”?

Figure 2 is quite hard to interpret. Would it be better if it were two figures? As is, it is quite confusing and doesn’t easily get the point across. An improvement in this figure would be helpful. Figure 3 and 4 are also quite messy.

Reviewer 2 ·

Basic reporting

See attached pdf

Experimental design

See attached pdf

Validity of the findings

See attached pdf

Additional comments

See attached pdf

Annotated reviews are not available for download in order to protect the identity of reviewers who chose to remain anonymous.

---

## Round 0.2 · accepted · Accept

Your revised manuscript has addressed the previous concerns expressed by the reviewers and is now suitable for publication.

#